# Skin Barrier Function: The Interplay of Physical, Chemical, and Immunologic Properties

**DOI:** 10.3390/cells12232745

**Published:** 2023-11-30

**Authors:** Paola Baker, Christina Huang, Rakan Radi, Samara B. Moll, Emmanuela Jules, Jack L. Arbiser

**Affiliations:** 1Department of Dermatology, Emory University School of Medicine, Atlanta, GA 30322, USA; pao_olaya@hotmail.com (P.B.); christina.huang@students.jefferson.edu (C.H.); rradi@emory.edu (R.R.); smoll@emory.edu (S.B.M.); ejules@emory.edu (E.J.); 2Sidney Kimmel Medical College, Thomas Jefferson University, Philadelphia, PA 19107, USA; 3Metroderm/United Derm Partners, 875 Johnson Ferry Road, Atlanta, GA 30342, USA

**Keywords:** skin barrier, acid mantle, psoriasis, atopic dermatitis

## Abstract

An intact barrier function of the skin is important in maintaining skin health. The regulation of the skin barrier depends on a multitude of molecular and immunological signaling pathways. By examining the regulation of a healthy skin barrier, including maintenance of the acid mantle and appropriate levels of ceramides, dermatologists can better formulate solutions to address issues that are related to a disrupted skin barrier. Conversely, by understanding specific skin barrier disruptions that are associated with specific conditions, such as atopic dermatitis or psoriasis, the development of new compounds could target signaling pathways to provide more effective relief for patients. We aim to review key factors mediating skin barrier regulation and inflammation, including skin acidity, interleukins, nuclear factor kappa B, and sirtuin 3. Furthermore, we will discuss current and emerging treatment options for skin barrier conditions.

## 1. Introduction

The skin is a key organ, serving as a chemical, physical, and immune barrier between the internal milieu and the external environment [1,2,3]. It serves as a permeability barrier to sustain terrestrial life, preventing excessive water loss, while protecting the body from various insults, from mechanical to microbial to oxidative. The outer layer of skin, the stratum corneum, is composed of anucleate corneocytes that are surrounded by lamellar bilayers composed of ceramides, fatty acids, and cholesterols, which all play roles in skin barrier function. For instance, the corneocytes contribute to mechanical integrity, mitigate ultraviolet radiation, and help regulate hydration, while the adjacent lamellar components form the permeability barrier, have antimicrobial properties, and counter free radical oxidation [4]. In conjunction with these protective properties, the components of the skin barrier also contribute to downstream signaling (for instance, initiating an inflammatory response), and elements of the skin barrier can influence one another [5]. As such, proper regulation of skin barrier components is essential, as dysregulation is implicated in multiple dermatologic conditions [6,7]. Previous reviews have covered multiple aspects of the skin barrier, including the biomolecular components, diverse functions, therapeutic considerations, and clinical significance, while this review emphasizes mechanisms of cross-regulation between multiple physical, chemical, and immunological processes and the resulting clinical implications [4,8,9,10].

This review discusses the current understanding of molecular and immunological regulation of the skin barrier and draws clinical correlations between an impaired skin barrier and associated skin conditions. It also proposes potential treatments that focus on maintaining the acidic pH and mitochondrial function of the skin, as well as limiting the activation of inflammatory pathways. In addition, this review seeks to highlight the interplay between the skin’s acid mantle, the role of ceramides, and immune regulation. These combined factors may help restore the homeostasis of the skin barrier and subsequently improve certain skin conditions.

## 2. The Roles of Acid, Sphingolipids, and Mitochondria in the Skin Barrier

The epidermis, the outer layer of the skin, acts as a key permeability barrier. It continuously renews itself, requiring strict regulation of proliferation and differentiation. The skin barrier function relies greatly on a variety of variables including the stratum corneum (SC), tight junctions, and immunologic surveillance by Langerhans cells.

In 1928, Dr. Alfred Marchionini first proposed the term “acid mantle” to describe the inherent acidic nature of the SC [11]. Under physiological conditions, the human skin is covered with a very thin layer of acid, rendering an acidic skin pH, while the internal body maintains a near-neutral pH [12,13]. This acidic pH of the skin influences the antimicrobial defense, barrier homeostasis, SC integrity, and function of several enzymes that are involved in synthesis and the maintenance of a healthy skin barrier. However, there are “physiologic gaps” in the epidermal acid barrier depending on factors that modify skin pH including the anatomical site, phototype, and age. For instance, intertriginous areas, lighter phototypes, and the skin of newborns and the elderly have a more alkaline pH [13,14,15]. This can disturb barrier function and favor colonization by pathogenic flora such as *S. aureus* and *C. albicans* [13,16,17].

Many factors contribute to the formation of this acid mantle. The activity of the sodium–hydrogen exchanger isoform-1 protein (NHE1) directly increases the proton concentration of the stratum corneum [18]. Other acidic factors, such as lactate and free fatty acids converted from phospholipids by secretory phospholipase A2 contribute to acidity [19,20]. It has also been proposed that filaggrin degradation products such as urocanic acid and pyrrolidine carboxylic acid contribute to skin acidity. Some studies have proposed that urocanic acid is responsible for the acid mantle of the skin, although this point has been contested [21,22,23]. Meanwhile, pyrrolidine carboxylic acid could also provide natural moisturizing factors in addition to its inherent acidity, further contributing to skin barrier protection [24,25,26,27].

Skin pH influences the bacterial flora of the skin. A normal skin flora includes coagulase-negative staphylococci that grow at an acidic pH. Concomitant, commensal bacteria on the epidermis along with a healthy skin barrier favor a low pH and the tonic secretion of defensive mediators such as IL-12, while an impaired barrier leads to an elevated pH. Conversely, pathogenic microorganisms such as *S. aureus* grow optimally at a neutral pH, which is clinically significant, as colonization with *S. aureus* is associated with atopic dermatitis and subsequent dysregulation of the skin barrier [12,13]. Alkaline pH could also contribute to the antibiotic resistance of methicillin-resistant *Staphylococcus aureus* (MRSA) to certain beta-lactams, as culture and phagolysosome studies have shown that pH < 5.5 promotes susceptibility to meropenem and cloxacillin in MRSA [28]. Additionally, when the skin is repetitively washed with alkaline soaps, there is a rise in the pH that favors the increase in the population of *C. acnes* and *S. aureus* [12,13].

Alterations in skin surface pH also influence enzymatic activity. An elevated pH is known to increase the activity of serine proteases including kallikrein 5 and kallikrein 7, which cause cytokine activation and inflammation, potentially through the protease-activated receptor 2-thymic stromal lymphopoietin pathway [29,30]. These enzymes contribute to the desquamation and degradation of corneodesmosomes by degrading desmoglein 1 [13,31], which leads to reduced SC cohesion, which is associated with decreased lamellar body secretion [13,32,33,34]. The loss of kallikrein inhibition via decreases in serine protease inhibitor Kazal type-5 (SPINK5) leads to Netherton syndrome, in which patients demonstrate impaired barrier function with epidermal hyperplasia and symptoms of atopic dermatitis [35]. As such, upregulation of kallikreins through changes in the skin surface pH could lead to multiple clinical symptoms. Additionally, regulatory enzymes including β-glucocerebrosidase and acid sphingomyelinase, which have an optimal pH of 5.6 and 4.5, respectively, require an acidic pH for proper function [13,36]. The ceramide levels in the SC are regulated by the balance of β-glucocerebrosidase, acid sphingomyelinase, and acid ceramidase, so any disturbance in these enzymes will alter the ceramide levels in the SC [13,37].

SC lipids such as ceramides also contribute to the barrier function of the epidermis, while also facilitating maintenance of an acid mantle. Stratum corneum lipids form a hydrophobic layer which, along with the tight junctions and desmosomes underlying the SC, prevent dehydration and retain water inside the skin [1]. Ceramides, a derivative of sphingolipids, are a marker of SC lipids. Ceramides are composed of an acyl chain and a sphingoid base. The barrier function of the SC is determined by the different subclasses of ceramides, which can differ in the acyl chain, sphingoid base structure, length of sidechains, and other structural variations [37,38,39]. As lipids, ceramides constitute part of the physical skin barrier, contributing to the intercellular lamellar sheets in the SC [40,41,42,43]. In this structure, ceramides help skin barrier health by repelling water, thus maintaining skin moisture by preventing water loss while also preventing skin irritation from outside substances [44]. Additionally, ceramides promote mitophagy, eliminating abnormal mitochondria [45]. Maintaining a healthy skin barrier requires intact mitochondria in all layers of the skin that produce the energy needed to pump out acid and maintain the acid mantle of the skin.

Ceramide deficiency may also be attributed to elevated activity of certain enzymes that exhibit increased activity in an alkaline environment, such as alkaline ceramidase, which is involved in barrier lipid degradation [13,36]. Studies have shown that pH elevation in normal skin alters the barrier function due to increased serine proteases activity and a reduction in ceramide precursor enzyme activity (5,6). Ceramides may also play a role in the immune regulation of the skin, as ceramides contribute to IL-12 induction, which then activates the Th1 immune program [46]. Overall, the interplay between acid, mitochondria, and ceramides is responsible for multiple factors related to skin health. (Figure 1). The roles of these three factors will also be discussed more in depth in the following sections in the context of skin barrier restoration. 

## 3. Immunology in Skin Barrier Function

The skin is often referred to as the body’s first line of defense, as it prevents entry of pathogens, and immune responses in the skin can eliminate many intruders [47]. However, aberrant overactivation of the immune system can be detrimental to the skin barrier function. Dysregulation of immune system programs is implicated in atopic dermatitis and psoriasis, which are diseases of impaired skin barrier function [48,49]. Helper T cells produce cytokine profiles that can be divided into Th1 and Th2. These enact different responses. The Th1 program, stimulated by IL-12, produces INF-gamma, IL-2, and tumor necrosis factor alpha/beta (TNF), leading to a cell-mediated immune response [50,51]. Th2 activation favors the production of IL-4, IL-5, IL-6, IL-10, and IL-13, inducing a more humoral response with production of antibodies (while also inhibiting macrophage functions) [51].

Traditionally, Th1 has been associated with inflammation, while Th2 cytokines counterbalance the Th1 response to avoid excessive inflammation [52]. Th1 has been implicated in psoriasis, although evidence suggests that Th17 expression could be more important in psoriasis, producing various proinflammatory factors including IL-17 and IL-22 [49]. Indeed, IL-12 expression has been observed to possibly reduce the expression of Th17 in psoriatic lesions [53]. In addition, some conditions with chronic inflammation such as atopic dermatitis display increased profiles of Th2 cytokines, implicating Th2 activity as a disruptor of normal tissue function [48]. In fact, Th2 activity is seen as a major driver in the development of atopic dermatitis [54]. Studies on atopic dermatitis found that the presence of Th2 cytokines (IL-4 and IL-6) at a concentration of 10 nM has been shown to reduce ceramides in the skin, a key component of a functional skin barrier [55]. The opposite effect is observed with the addition of Th1 cytokines, demonstrating a possible role for Th1 in maintaining the skin barrier in models of atopic dermatitis [55].

Th1 inhibits the Th2 response [56]. Because of this effect, compounds such as S12 and S14 (derivatives of solenopsin, a lipophilic alkaloid found in fire ant venom) that stimulate Th1 activity through the activation of IL-12 could be beneficial for skin barrier homeostasis [57]. Further research into understanding the balance between Th1 and Th2 responses regarding the skin barrier could be helpful, given that Th1 is more heavily implicated in psoriasis, and Th2 is more implicated in atopic dermatitis, although they are both disorders that are characterized by a disrupted skin barrier. Furthermore, more investigation into activating Th1 could be helpful, given its proinflammatory effects, but if Th17 is more predominant than Th1 in driving psoriasis, the drawbacks of Th1 activation could be outweighed by the possible benefits in restoring the skin barrier. Through IL-12 induction, the solenopsin analogs S12 and S14 favor IL-12-driven Th1 expression, consequently suppressing the Th2- and Th17-mediated disruption of the skin barrier [57].

## 4. Significant Factors in the Disruption and Restoration of the Skin Barrier

Dysfunction of the skin barrier is a hallmark of multiple dermatologic conditions, including psoriasis and atopic dermatitis [6,7]. Many treatments, including certain formulations of emollients, contribute to the restoration of the barrier [58,59]. Emollients with biochemical properties, such as ceramide-based mixtures, have had success in improving atopic dermatitis and psoriasis [60,61]. Aside from helping maintain a physical skin barrier, ceramides also demonstrate a biochemical signaling component that is relevant to the skin barrier, influencing cellular pathways including those associated with cell differentiation [62].

Ceramides also stimulate the Th1 immune response that is mediated by IL-12, which plays a role in downregulating Th2- and Th17-driven inflammation [46,53,56]. The depletion of ceramides has also been shown to induce Th2 inflammatory responses in mouse models [63]. Ceramide-induced mitophagy contributes to an intact skin barrier, as dysfunctional mitochondria induce metabolic issues including excessive production of superoxide, which could kill cells that are needed to maintain a skin barrier [64]. Ceramide can be converted to sphingomyelin, which can be further metabolized to sphingosine-1-phosphate (S1P) [65]. Sphingosine-1-phosphate has demonstrated signaling, which controls proliferation and induces differentiation in keratinocytes, opposing the undifferentiated, uncontrolled keratinocyte proliferation that is characteristic of psoriasis [66]. Thus, the disruption of ceramides may lead to skin barrier disruption.

However, sphingosine-1-phosphate has also been linked with inflammation through immune cell recruitment and angiogenesis, which could be disruptive to a healthy skin barrier [67]. Accordingly, ceramide analogs, which cannot be converted to S1P, have been synthesized as derivatives of solenopsin. These analogs, S12 and S14, result in a reduction in thickness of inflammatory lesions in mouse models, along with a reduced penetrance of CD4+, CD8+, and CD11c+ T cells. They have also demonstrated various biochemical signaling influences that highlight the role of biochemical pathways in skin barrier maintenance [57]. These include downregulation of IL-22, upregulation of IL-12, and decreases in immune cell recruitment [57]. By examining the effects on the skin barrier both in a physical and biochemical manner, further studies could use this dual-pronged concept to develop treatments with greater efficacy in restoring the skin barrier.

Mitochondrial health also plays a large role in maintaining a healthy skin barrier. Skin cells with sufficient energy produced by healthy mitochondria maintain a barrier, while senescent cells could have mitochondrial dysfunction, which disrupts barrier function [68]. First, mitochondria generate ATP, which is used by the sodium–hydrogen exchanger isoform-1 protein, which helps export protons and decrease extracellular pH, contributing to skin acidity [18,69]. Furthermore, cellular aging, associated with mitochondrial dysfunction, leads to other adverse effects that decrease the skin barrier effectiveness, including reduced rete ridges [68,70]. Signals for cell senescence including elevated reactive oxygen/radicals from defective mitochondria can increase the expression of NFκB, which can lead to hyperproliferation of keratinocytes, leading to disrupted barrier function [71,72,73]. Furthermore, senescent cells can have adverse effects on the skin barrier by affecting the microenvironment: the senescence-associated secretory phenotype (SASP) produces many proinflammatory cytokines [74]. Mitochondrial function could possibly be preserved with SIRT3 activators, including honokiol, a small molecule polyphenol with anti-inflammatory properties, thus contributing to normal skin barrier function [75] (Figure 2).

Targeting mitochondria could also alleviate oxidative stress and subsequently restore barrier function. Increased levels of mitochondrial oxidative stress, with elevated superoxide dismutase 2 and hydrogen peroxide, were observed in mouse models of atopic dermatitis, and a filaggrin deficiency may have contributed to this oxidative profile. The subsequent treatment with mitochondrial-targeting antioxidants reduced inflammation and molecular damage, with restoration of epidermal homeostasis [76].

CD4+, CD8+, and CD11c+ immune cells have all been implicated in inflammatory responses that are detrimental to the skin barrier [77,78]. In 3-D models of skin, T cell infiltration induced hyperproliferation of keratinocytes and delayed epidermal differentiation, some of the key hallmarks of psoriasis [79]. The production of IL-17A and IL-22, key inflammatory cytokines that are heavily implicated in the pathogenic of psoriasis, by CD4+ T cells has marked CD4+ T cells as a main target of research. More recently, CD8+ cells have been also recognized to produce these cytokines in both psoriasis and atopic dermatitis, highlighting the importance of reducing both CD4+ and CD8+ infiltration in skin lesions [77,80]. Indeed, studies have observed that psoriatic skin has T cells that mainly express CD8+, which subsequently produces IL-17, a cytokine that drives psoriasis development. Consequently, targeting these cells has prevented psoriasis in vivo [77]. CD11c+ dendritic cells have also been observed to be highly concentrated in psoriatic skin [78]. Methotrexate, a prominent treatment for psoriasis, hinders T cell infiltration into the skin [81]. Similarly, S12 and S14 reduce the infiltration of CD4+, CD8+, and CD11c+ cells in mouse models of psoriasis [57].

Vascular changes are also associated with skin barrier disruption, as a sufficient oxygen supply of the epidermis is required for skin barrier health. Previous mouse studies indicate that acute insults to skin barrier stimulates vascular endothelial growth factor-A (VEGF-A) expression to promote capillary growth in the papillary dermis. Knockdown of VEGF was associated with abnormal permeability barrier homeostasis, possibly due to a decreased epidermal lamellar body production and reduced vascularization [82]. Conversely, excessive VEGF production is also associated with psoriasis (which may result from trauma, as seen in the Koebner phenomenon), which is consistent with the observation of abnormal papillary angiogenesis in psoriasis [82,83]. Evidence of increased angiogenesis is also seen in inflammatory skin regions that are affected by atopic dermatitis, and mast cells may induce angiogenesis through the release of VEGF-A and VEGF-B [84].

## 5. Interleukins and Toll-like Receptors in Skin Barrier Regulation

Multiple interleukins, such as IL-22, play a significant role in the regulation of the skin barrier [85]. IL-22 is part of the IL-10 cytokine family, is primarily produced by CD4+ T cells, and signals through the IL-22R heterodimeric receptor [85,86,87]. Although IL-22 mainly reinforces skin barrier function by promoting inflammation, continuous, unregulated IL-22 expression causes unwanted inflammation [86]. This exaggerated inflammatory response further contributes to psoriasis and atopic dermatitis [86,87]. Studies comparing patients afflicted with psoriasis to a control group show a significantly higher IL-22 expression in the psoriasis group [88]. Furthermore, IL-22 activity inhibits keratinocyte differentiation, contributing to the presentation of undifferentiated cells that is seen in psoriatic skin [89,90,91]. IL-22 also induces hyperproliferation of keratinocytes, further contributing to psoriatic skin lesions [92]. Additionally, mouse models show that the activation of IL-22 results in atopic dermatitis through an increased presence of CD4+ and CD8+ T cells, which induce inflammation [87]. Over-expression of IL-22 is therefore disruptive to normal skin barrier function. Accordingly, recent data suggest that targeting IL-22 may prove effective in treating psoriasis and other inflammatory conditions [88]. Solenopsin analogs S12 and S14 have been shown to downregulate IL-22, decreasing epidermal thickness and CD4+/CD8+ T cell infiltration [57].

IL-17 also plays a significant role in the disruption of a normal skin barrier. IL-17 is a proinflammatory cytokine that is expressed heavily in psoriatic skin [93]. IL-17 promotes inflammation by contributing to the buildup of neutrophils and by upregulating chemokines that are known to cause psoriasis [93]. Further, IL-17 has been found to downregulate IL-12, which is responsible for inducing IFN-gamma expression in Th1 cells [94]. Th1 cells play a crucial role in the cell immune response and protection from pathogens [95]. Therefore, by suppressing IL-12 expression, IL-17 inhibits both IFN-gamma expression and, subsequently, Th1 expression [94]. These conclusions, along with a recent trial showing promising results after reducing the IL-17 expression in patients with psoriasis, suggest that inhibiting the IL-17 family may effectively target psoriasis [93].

Although it is primary a driver of psoriasis, IL-17 assumes a more minor role in promoting atopic dermatitis [96]. IL-17 is expressed in atopic dermatitis lesions in relatively low concentrations [96]. Interestingly, IL-17′s receptor A (IL-17RA) seems to be positively correlated with skin barrier function. In mouse models of atopic dermatitis, IL-17RA absence not only worsened, but also spontaneously caused inflammation [97]. It was concluded that mice lacking IL-17RA have overall defective barriers with dysbiotic skin microbiomes [97]. Furthermore, this inflammation appeared to be mediated by a Th2 immune response and improved upon antibiotic treatment, showing that IL-17RA expression could play a significant role in modulating an appropriate immune response and maintaining a healthy skin barrier [97]. Recent treatments that target IL-17, including brodalumab, ixekizumab, and secukinumab, have shown clinical efficacy in the treatment of skin barrier disorders including psoriasis [98].

IL-12 is a key interleukin that was previously targeted as a treatment for psoriasis (with antibodies targeting both IL-12 and IL-23) [99]. However, more recent research has suggested a restorative role in the skin barrier for IL-12 [53,100]. IL-12 was the first member that was discovered from the IL-12 cytokine family [101]. It has been implicated as a proinflammatory cytokine that promotes the production of IFN-gamma and stimulates the Th1 immune response [101,102]. Despite being seen largely as a proinflammatory cytokine, there are certain contexts wherein IL-12 may limit inflammation and offer other benefits to skin barrier restoration [53,57]. After stimulus impacting both the physical and immunological aspects of the skin barrier, as seen in radiation damage, IL-12 therapy was able to rescue aspects of the damaged skin. This included reduced water loss (indicating a physical barrier restoration) and an increased density of interstitial dendritic cells (indicating enhanced immunological barrier maintenance) compared with skin that was irradiated without IL-12 treatment [100].

Furthermore, IL-12 signaling has been reported to limit skin inflammation in psoriatic lesions [53]. Although IL-12/23 has been targeted by therapies to reduce inflammations in previous studies, more recent data suggest that IL-23, which shares the p40 subunit with IL-12, may play a greater role in inflammatory responses, while IL-12 exhibits some anti-inflammatory activity [53,103,104,105,106]. Because of this, treatments targeted at p40 inhibition, such as ustekinumab, may have efficacy by inhibiting IL-23 [107]. Indeed, IL-23 expression is enhanced in keratinocytes in psoriasis lesions compared with keratinocytes from healthy skin [108]. However, this also inhibits IL-12, which could render the treatment less effective than an IL-23-specific treatment [53]. IL-12 also helps inhibit the Th17 immune response, which can drive psoriasis through the production of IL-17 [92]. IL-12 has been shown to inhibit Th17 cells in the context of pulmonary disease through IL-10 signaling [109]. In the skin, IL-12 expression can induce stromal alterations, which can reduce Th17 levels [53]. S12 and S14, which have been shown to effectively restore the skin barrier, increase IL-12 expression in mouse models of psoriasis [57]. S14 also reduced the Th2 response in a mouse model of atopic dermatitis, with a decrease in the IL-4 expression associated with increased IL-12 [110]. Furthermore, solenopsin increases IL-12, possibly independently of upregulating the shared p40 subunit, as it does not influence IL-23 levels [57].

IL-1 family cytokines also mediate inflammation and atopy, with studies implicating the dysregulation of IL-1α, IL-1β, IL-18, IL-33, IL-36α, IL-36β, and IL-36γ in different processes of skin barrier disruption [111]. Keratinocyte IL-1α secretion in filaggrin-deficient mouse models perpetuate chronic inflammation, while IL-1β signaling in mouse models induces IL-17 expression, indicating a role in psoriasis [111,112]. Neutrophils proteases, which may be found in higher concentration in psoriatic skin, truncate IL-1α/β, IL-33, and IL-36α/β/γ, which could further amplify inflammatory responses [113]. IL-33, which induces atopic dermatitis-like inflammation in mouse skin and is overexpressed in atopic dermatitis skin samples, can also be activated by environmental proteases, potentially exacerbating the response in atopic dermatitis [114,115]. IL-18 induction using *S. aureus* in mouse models of atopic dermatitis may also mediate the inflammatory response [116]. IL-36α and IL-36γ injection into wild-type mice led to an increased skin expression of IL-17 and IL-23, implicated in psoriasis, while the latter also induced neutrophilic infiltration. These inflammatory effects were also inhibited by the IL-36 receptor blocking antibodies [117]. Notably, elevated IL-36 cytokine pathway activity is implicated in the pathogenesis of generalized pustular psoriasis (GPP), as up to 1/3 of GPP patients have missense mutations in IL-36RN, which inhibits IL-36 receptor activation [117]. Other mutations in GPP patients, such as in CARD14, AP1S3, SERPINA3, and MPO, also regulate the IL-36 signaling axis [118]. Recent anti IL-36 receptor antibodies, namely spesolimab, have been FDA-approved for treatment of generalized pustular psoriasis [119].

Toll-like receptor 4 (TLR4) is another important regulatory factor that must be balanced to ensure an appropriate immune response. Overactivation of TLR4 is observed in psoriasis [120]. In mouse models, TLR4 both helps to initiate the development of psoriasis plaques and to maintain the presence of these lesions [120]. When neutrophils infiltrate the epidermis in psoriatic lesions, they release neutrophil extracellular traps, which subsequently cause inflammation through TLR4 signaling [121]. In addition, TLR4 could interact with TLR2, leading to an autoimmune response. Consequently, inhibition of TLR4 has reduced psoriatic symptoms in mouse models [121]. Solenopsin analogs, which have shown success in mouse models of psoriasis, also downregulate TLR4 expression [57]. Furthermore, studies of TLR4 expression in healthy skin versus skin with atopic dermatitis, contact dermatitis, and psoriasis found that TLR4 became more expressed in the upper layers of skin with these conditions compared with healthy skin, where TLR4 was mainly in the basal layers [122]. Interestingly, a decreased TLR4 expression is detrimental to patients with atopic dermatitis, as a decreased expression disrupts the water barrier and increases skin thickness [123]. This benefit occurs both within and outside the context of pathogen infection, as even without infection, TLR4-deficient mice displayed more severe atopic dermatitis with a stronger T helper 2-driven immune response compared with wild-type mice [124]. TLR4 may be beneficially activated in some of these conditions, while over-activation can lead to inflammation [122].

## 6. Clinical Applications

The homeostasis of the skin barrier is disrupted in many inflammatory skin diseases, and immunological dysfunction can cause skin barrier disturbances [1,125]. Virtually all inflammatory and malignant skin disorders arise from chronic barrier defects. Additionally, defects in the skin barrier function due to genetic mutations or mechanical stimuli such as scratching, which can trigger inflammatory responses [1].

Aging has an impact on the barrier function, characterized by epidermal aging (decreased mitochondria, ATP, acidification, and antimicrobial peptides) and dermal aging (decreased mitochondria, ATP, and collagen alongside increased NFκB).

Inflammatory skin conditions that are characterized by a disruption of the barrier function result in the alkalinization of the skin barrier. Examples include atopic dermatitis, contact dermatitis, psoriasis, and acne [13]. Atopic dermatitis and contact dermatitis are Th2-mediated, and psoriasis and acne are Th17-mediated. In general, skin inflammatory conditions with barrier dysfunction can be classified into two groups: one that is associated with the expression of antimicrobial peptides (AMPs) such as psoriasis and the other without the expression of AMPs such as atopic dermatitis and eczema [126].

### 6.1. Psoriasis

Studies have shown that the application of polyhydroxy acids improves barrier function in neonatal and aged rodent skin and super-normalize the barrier function in normal mice and humans [127,128,129]. Given that the function of enzymes that are involved in desquamation is pH-dependent, the use of acidic preparations is beneficial to promoting keratolysis [13].

A study has shown that palmoplantar psoriasis has an excellent response to the combined treatment with Trichloroacetic acid (TCA) 40% peels and gentian violet (GV), as TCA helps restore the normal acidity of non-inflamed skin [130]. The benefits of GV on inflamed skin include the eradication of pathogenic Gram-positive bacteria, bactericidal activity against dermatophytes and mold, decreased vascular permeability, improved efficacy of steroids, and anti-inflammatory activity through inhibition of nicotinamide adenine dinucleotide phosphate oxidase, decreasing inflammatory cytokines like Angiopoietin-2 [130,131,132].

### 6.2. Atopic Dermatitis

Studies have shown that the pH in eczematous skin and uninvolved skin of children with atopic dermatitis is higher in comparison to the skin of healthy children [13,133,134]. Free amino acids and urocanic acid, which are involved in creating the acid mantle, are reduced in atopic skin. When combined with a filaggrin mutation, this provides a danger signal to the keratinocyte nucleus [13,21]. Sweat secretions that are rich in lactic acid, which are also thought to contribute to the acid mantle, are reduced in atopic skin [13]. The impaired secretion of lamellar bodies seen in atopic dermatitis may increase pH, as this process is a source of protons for SC acidification [13].

This alkalization of the skin and changes to the skin barrier can increase the chance of pathogenic colonization. Growth of *S. aureus* colonization is optimal at a neutral pH and markedly inhibited at pH values around 5 [13,135]. Studies have shown that the skin of patients with atopic dermatitis have a decreased ability to produce AMPs including cathelicidin and β-defensins-2 and -3 [126]. Barrier defects permit an increased *S. aureus* entry and subsequent expression of Th2 cytokines and IL-17 and a decreased expression of cathelicidin. This is consistent with studies showing that Th2 cytokines downregulate the induction of cathelicidin in the skin [126,136]. Cathelicidin LL37 has been found to exhibit effective antimicrobial activity against *S. aureus* [137].

Skin barrier changes can also induce symptoms, which may be alleviated by corrections in pH. Serine proteases, which have increased function at an elevated pH, induce pruritus via PAR-2 receptor activation in keratinocytes and nerves in atopic skin [138]. Moreover, studies on hapten-induced atopic dermatitis in at-risk mice show that lowering the pH reduces the Th2 inflammatory response, prevents epidermal hyperplasia, reduces tissue eosinophilia, and normalizes the epidermal structure [13].

### 6.3. Contact Dermatitis

Individuals that are prone to irritant contact dermatitis have demonstrated higher SC pH values compared with healthy individuals (69–70). An alkaline pH induces an alteration in the SC integrity, with the consequent disturbance of the barrier homeostasis that makes the skin susceptible to injury from external substances and mechanical forces [13,32].

Topical alpha-hydroxy acids such as lactic acid are used in treating disorders of keratinization. These acids increase keratinocyte ceramide production, leading to an improved barrier function. This also reduces the sensitivity to sodium lauryl sulfate in susceptible individuals [13,139]. Therefore, the use of these acids may be beneficial in individuals with reduced barrier function, such as atopic patients who have reduced levels of ceramides and patients that are prone to irritant contact dermatitis. Other studies have shown the benefits of topical acidic electrolyte water (pH 2.0–2.7) on *S. aureus* colonization in patients with atopic dermatitis [13,140].

## 7. Conclusions

Continued investigations on the nature of the skin barrier and the multiple factors that are involved in its regulation will help in the treatment of common skin disorders. The restoration of the acid mantle of the skin can help treat inflammatory skin diseases, in part by reducing inflammatory cytokines.

The acid mantle is required for maintaining the barrier function, because decreased acidity leads to inflammation and infection. The skin has a tonic level of Th1 inflammation, which is maintained by IL-12. IL-12 suppresses Th2-mediated immunity that is involved in atopic dermatitis and the Th17-mediated immunity that is involved in psoriasis. Additionally, a combination of skin aging and external factors results in inflammation.

The treatment of inflammatory skin diseases has evolved. Initially, lymphocytes were targeted with steroids, methotrexate, cyclosporine, and Janus kinase inhibitors, with potential systemic toxicity and carcinogenesis being associated with these therapies. Currently, cytokines made by keratinocytes are being targeted with biologics such as TNF inhibitors and anti-IL-17, -IL-23, -IL-4, and -IL-22 antibodies. In the future, small molecules may stop keratinocytes from secreting excess cytokines. Treatments may include NFκB inhibitors, proteasome inhibitors, and SIRT3 activators to decrease aging-related cytokines. Furthermore, GV, TCA, and new compounds such as ceramide-based lipids may be key agents in restoring skin barrier homeostasis and improving inflammatory skin conditions. Ceramide restoration by itself might not help if the ceramides are converted to S1P. Derivatives of solenopsin, including S12 and S14, which cannot be converted to S1P, may be more effective in restoring the skin barrier. Ceramide analogs also induce Th1 signaling, downregulating Th2 and Th17, and therefore restoring the skin immune balance, which is necessary to improve inflammatory cutaneous conditions. 

## Figures and Tables

**Figure 1 cells-12-02745-f001:**
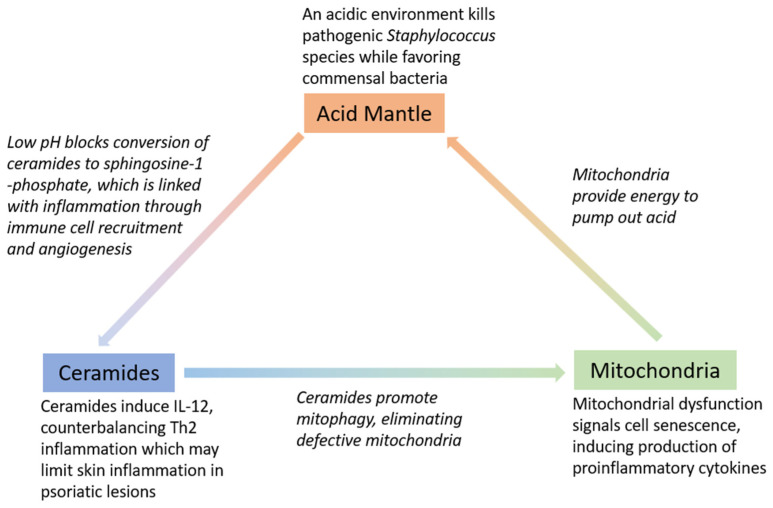
Interplay between acid, ceramides, and mitochondria and key roles in maintaining skin barrier.

**Figure 2 cells-12-02745-f002:**
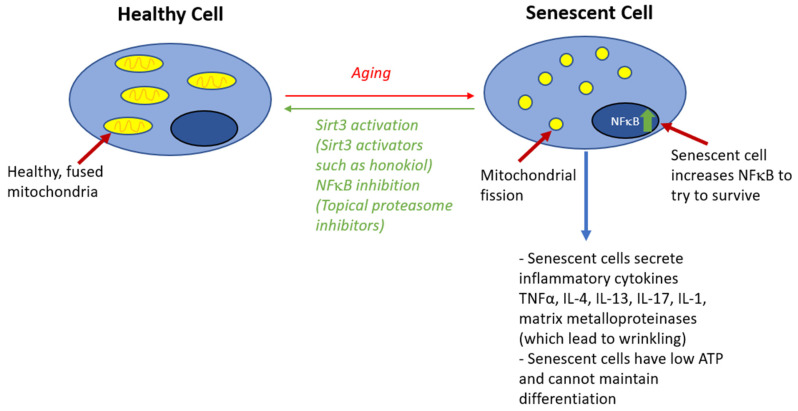
Role of cell senescence signaling in disruption of skin barrier.

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
