# Peer review of "Skin Barrier Function: The Interplay of Physical, Chemical, and Immunologic Properties"

_cells, 2023, doi:10.3390/cells12232745_

Round 1

Reviewer 1 Report

Comments and Suggestions for Authors

This is an outstanding and timely review about skin barrier function and skin pathophysiology. The authors did a wonderful job brining all the basic concept of skin biology, integration with other organ system including immunology and finally discuss about the pathology. Brining ceramides and mitochondria dysfunction with the influence with environments parameters including acid, is a great addition of the review. 

I have only one suggestion is that addition of vascular function and pathogenesis can play a role in this contest would improve the article very much. Again, addition of a paragraph will suffice considering the expertise of the authors.  

Comments on the Quality of English Language

Little restructuring of some of the wording might help for the general readers. 

Reviewer 2 Report

Comments and Suggestions for Authors

The review manuscript by Paola Baker and colleagues discusses the roles of the acidic pH for the barrier function of the skin in health and diseases. This is a timely and interesting review. I have a few comments.

In contrast to the title, this review covers only specific topics. Citation of other reviews may help the reader to obtain information about other skin barrier topics and put this review into a broader context.

Lines 63-65: “Urocanic acid acts as a chromophore for ultraviolet B radiation absorption and is responsible for the acid mantle of the skin.” There is no proof for the impact of urocanic acid on the skin pH, although it has been discussed in some papers and therefore can be presented as hypothesis. Mice that are not able to produce urocanic acid due to a genetic defect of histidase were reported to have apparently normal epidermis, apart from increased sensitivity to UV radiation.

Line 74: Why “Methicillin Resistant Staphylococcus Aureus (MRAS)”? Any link between alkaline pH and methicillin? Please check abbreviations and names of bacteria such as “Streptocuccus”.

Line 132, “Traditionally, Th1 is associated with inflammation while Th2 counters this response.” What is the basis for this statement?

Line 144: What is S12?

Line 145: What is S14?

Line 198: What is honokiol? The description as a “compound” is not informative.

Section 5 (Interleukins and Toll Like Receptors in Skin Barrier Regulation) lacks important information on the interleukin-1 family. Some members of this family, such as IL-36RN, are expressed before the cornification of epidermal keratinocytes, and IL36RN gene mutations cause diseases, such as generalized pustular psoriasis.

As figures 1 and 2 highlight the role of mitochondria in the epidermis, the authors may consider additional references to support this concept. Recently, a paper linked mitochondria, filaggrin and lipids:

Minzaghi et al. Excessive Production of Hydrogen Peroxide in Mitochondria Contributes to Atopic Dermatitis. J Invest Dermatol. 2023;143(10):1906-1918.e8. doi: 10.1016/j.jid.2023.03.1680.  

NF-kappa B is not spelled correctly several times in the pdf manuscript

The format of the references is unusual and not consistent: sometimes complete journal names, sometimes abbreviations. The authors should carefully check the format of the references in the final version of the manuscript.

Conflicts of Interest, “JLA is the co-inventor of Patent US9592226B2”: First, JLA does not fit to any author on the title page. The list of authors or this statement needs to be corrected. Second, patent US9592226B2 is not mentioned in the text and its claims are not described in the Conflicts statement. Information about this patent must be provided to inform the reader about which parts of the paper are affected by the conflict of interest.

Comments on the Quality of English Language

The quality of English language is good.
